# SRHand: Super-Resolving Hand Images and 3D Shapes via View/Pose-aware Neural Image Representations and Explicit 3D Meshes

**Minje Kim**    **Tae-Kyun Kim**
School of Computing, KAIST
{minjekim, kimtaekyun}@kaist.ac.kr

## Abstract

Reconstructing detailed hand avatars plays a crucial role in various applications. While prior works have focused on capturing high-fidelity hand geometry, they heavily rely on high-resolution multi-view image inputs and struggle to generalize on low-resolution images. Multi-view image super-resolution methods have been proposed to enforce 3D view consistency. These methods, however, are limited to static objects/scenes with fixed resolutions and are not applicable to articulated deformable hands. In this paper, we propose SRHand (Super-Resolution Hand), the method for reconstructing detailed 3D geometry as well as textured images of hands from low-resolution images. SRHand leverages the advantages of implicit image representation with explicit hand meshes. Specifically, we introduce a geometric-aware implicit image function (GIIF) that learns detailed hand prior by upsampling the coarse input images. By jointly optimizing the implicit image function and explicit 3D hand shapes, our method preserves multi-view and pose consistency among upsampled hand images, and achieves fine-detailed 3D reconstruction (wrinkles, nails). In experiments using the InterHand2.6M and Goliath datasets, our method significantly outperforms state-of-the-art image upsampling methods adapted to hand datasets, and 3D hand reconstruction methods, quantitatively and qualitatively. Project page: `https://yunminjin2.github.io/projects/srhand`.

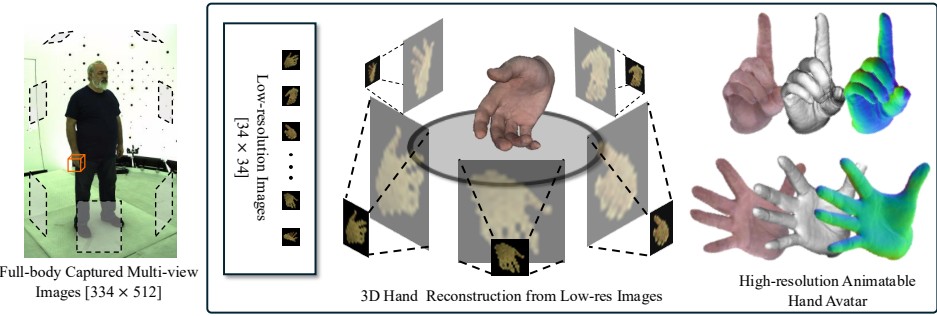

Figure 1: From low-resolution multi-view RGB images, SRHand reconstructs highly detailed animatable hand avatars by super-resolving both images and 3D geometry.

## 1 Introduction

Human avatars play a crucial role in immersive human-computer interaction within virtual and augmented reality (VR/AR) environments. To provide realistic user experiences, it is essential to reconstruct the full body with high fidelity, both in geometry and appearance [1, 14, 22]. Among

39th Conference on Neural Information Processing Systems (NeurIPS 2025).

these, hands are particularly challenging due to their fine-grained geometry, frequent occlusions, and high articulations. In typical full-body capture systems, the hand occupies less than 1.5% of captured images, making high-fidelity hand reconstruction from low-resolution (LR) regions a difficult task as illustrated in Fig. 1. In a multi-view setup, the resolution of the captured hand region varies with human poses and camera positions, making the reconstruction problem even more challenging.

While there are various studies [10, 39, 28, 27] on the hand domain, achieving expressive hand avatars is an important task for realistic human representation. S2Hand [7] and AMVUR [18] reconstruct on the MANO model, and they suffer from blurry textures caused by the low resolution of their hand meshes. This issue is tackled by increasing mesh vertex resolution for clearer textures and more accurate geometry. MANO-HD [5], HARP [20], UHM [39], and XHand [10] improve their hand model with surface subdivision, enabling to capture fine geometric details. However, they are heavily reliant on the high resolution input images from various viewpoints. LR images, shown in Fig. 1, lead to a substantial loss of details in 3D reconstruction.

Increasing the resolution of LR hand images has not been studied as a mainstream, up to our knowledge. Super-resolution (SR) of generic images instead has been extensively explored [49, 48, 11, 23], which aim to recover high-frequency details from LR inputs. However, hallucinated textures and multi-view inconsistencies when applied to individual images stem from being used for multi-view 3D reconstruction. Several works [26, 47, 57, 55, 9] have tackled SR from LR multi-view images enforcing 3D view consistency using NeRF [36] and GS [21], but they are limited to static objects, fixed SR scales, non-respective on 3D shapes, and not appliable to dynamic articulated targets like hands. Note variation of hand poses in frames makes the problem more challenging.

To address these limitations, we propose SRHand, a novel framework for super-resolving hand avatars in both 2D images and 3D shapes from limited-resolution multi-view/pose images. We tightly integrate an image super-resolution module and a 3D hand reconstruction module for high-fidelity avatar modeling. Considering the variation of captured hand region resolution, we propose a Geometric-aware Implicit Image Function (GIIF), which super-resolves hand images to arbitrary scales, while being conditioned on surface normals. Leveraging geometric-structured representation from explicit mesh-based reconstruction, we optimize the SR module ensuring consistency across varying view and pose images. This joint formulation removes blurred, overbounded shapes, and flickering artifacts and maintains accurate 3D structure across poses and viewpoints. We validate SRHand on real-world datasets, demonstrating significant improvements in appearance and geometry quality compared to existing SR methods adapted to hand images and 3D hand reconstruction baselines. The framework enables reconstructing hand avatars with wrinkles, nails, and subtle shape variations, even from low-resolution image inputs, which is essential for realistic, interactive VR/AR applications.

In summary, our main contributions are as follows:

- We introduce SRHand, a novel framework that super-resolves both 2D images and 3D shapes by integrating implicit image representations with explicit 3D meshes from LR images.
- We propose a geometric-aware implicit image function (GIIF), that conditions implicit neural representations on normal maps from a template model while leveraging adversarial learning to enhance texture fidelity.
- We jointly fine-tune the hand SR module with the 3D reconstruction process to enforce multi-view/pose consistency.

## 2 Related Work

**3D Avatar Reconstruction.** Human avatar reconstruction has been widely explored using various representations, including implicit fields such as neural radiance fields (NeRF) [36, 17, 53, 5], Gaussian Splatting (GS) [25, 16, 38, 44], and explicit meshes [56, 50, 52, 18, 24, 39, 8]. While NeRF and GS approaches excel in producing photo-realistic renderings, they often require dense viewpoints and struggle to represent accurate geometry. In contrast, mesh-based methods explicitly represent geometric structures but are highly dependent on 3D resolution for optimal representation.

Previous studies have achieved expressive hand [3, 20, 24, 39, 8, 10] avatars using explicit mesh-based methods. Specifically, several works [20, 24, 39, 8, 43, 3] have been proposed to represent hand appearance based on UV texture map rendering. UV texture rendering provides higher texture details,

even with low vertex resolutions. I2UV-HandNet [3] focuses on reconstructing a high-resolution UV-map hand mesh without texture from a single image. However, these approaches fall short of representing fine-grained 3D geometric details. In contrast, vertex color-based representations have been studied in [7, 18, 10]. Due to the fixed resolution constraint, S2Hand [7] and AMVUR [18] suffer from blurry textures caused by the low resolution of their hand mesh (778 vertices). Several works have been explored to upsample hand meshes. OHTA [62] utilizes MANO-HD (12,337 vertices) from HandAvatar [5], and URHand [8] relies on a mesh model proposed in UHM [39] (15,930 vertices). Following XHand [10], our mesh representation comprises over 49,000 vertices and 98,000 faces, enabling the capture of fine geometric details. In this way, our mesh representation narrows the performance gap with UV-based texture rendering methods while surpassing geometric-level details representation. However, XHand still falls short in capturing detailed geometric shapes on hand meshes since it does not consider whether the visual appearance is caused by texture colors or geometric shapes. Our mesh representations better model finer details by learning the geometric appearance and the texture color appearance. All aforementioned methods remain highly sensitive to input image resolution. LR images lead to degraded texture fidelity and loss of fine geometric detail, limiting performance.

**Image Super Resolution.** Super-resolution (SR) of generic images has been extensively studied. Since SR is a kind of generative method, there exists a distribution of possible SR results from LR images. While many research [49, 48, 11, 23, 2] have been proposed, we employ implicit neural representation methods as those methods are spatially adaptive to the 3D surface and capable of controlling image resolution per need while achieving plausible results. Local Implicit Image Function (LIIF) [6], which is the most representative work in 2D space, proposes to represent images in the continuous space, allowing it to generate arbitrary-resolution images. This approach eliminates the need for retraining when scaling to arbitrary magnification factors (*e.g.*, ×4, ×12.8), offering a unified solution for diverse super-resolution tasks. Building on implicit image functions, LIIF-GAN [19] introduces adversarial learning to enhance sharper textures and more realistic details. We enhance the fidelity of hand neural representations by leveraging normal maps derived from the MANO [45] template, while preserving realistic textures and fine-grained details through adversarial learning.

**Multi-view Super Resolution with 3D Consistency.** Previous works [31, 33, 34, 55, 32, 54, 58, 4, 46] propose to leverage generative 2D image prior using denoising diffusion [15] to reconstruct 3D objects from single or multi-view images. Especially in [55, 58, 32] multi-view consistency problems are addressed using confidence obtained from 3D space. Recent works [26, 55, 57, 9] have attempted to integrate image super-resolution techniques with NeRF for 3D view consistency. DiSR-NeRF [26] exploits a pre-trained diffusion model to reconstruct details that are missing in a low-resolution NeRF (LR NeRF) optimization. DiSR-NeRF is dependent on prompt guidance, otherwise the reconstructed details deviate from the ground truth. Yoon *et al.* [55] refines SR images by aggregating features from the SR module, NeRF representation, and uncertainty estimates of NeRF. While effective, this method relies on several pre-trained components which limit its adaptability. SuperNeRF [57] generates high-frequency details from LR NeRF consistency by searching the latent space of ESRGAN [49] for view-consistent solutions. Yet, enforcing multi-view consistency in the LR domain does not guarantee that obtained high-frequency details remain consistent across views. SRGS [9] utilizes the Gaussian splatting (GS) [21] method, which is known to be effective for rendering static scenes. However, GS are well known to fit will in static scenes and likely lose geometric details.

The aforementioned works have attempted to preserve multi-view consistency between super-resolved images for static scenes. None of these approaches has effectively tackled the challenge of preserving consistencies for dynamic, deformable objects, i.e. pose as well as view consistency. They are mainly for rendering novel view images, not offering explit 3D shapes. Besides, the required number of images dramatically increases for reconstructing animatable avatars, increasing computational costs and complexities. Our method overcomes these limitations by fine-tuning the SR module, enforcing multi-view and pose consistency while enhancing the fidelity of fine details in both images and 3D shapes, making our approach suitable for dynamic hand avatars.

## 3  Methods

We propose SRHand, a method for reconstructing a detailed hand avatar in both 2D and 3D representation. Fig. 2 shows the overall framework of SRHand. Sec. 3.1 explains the SR module which is a

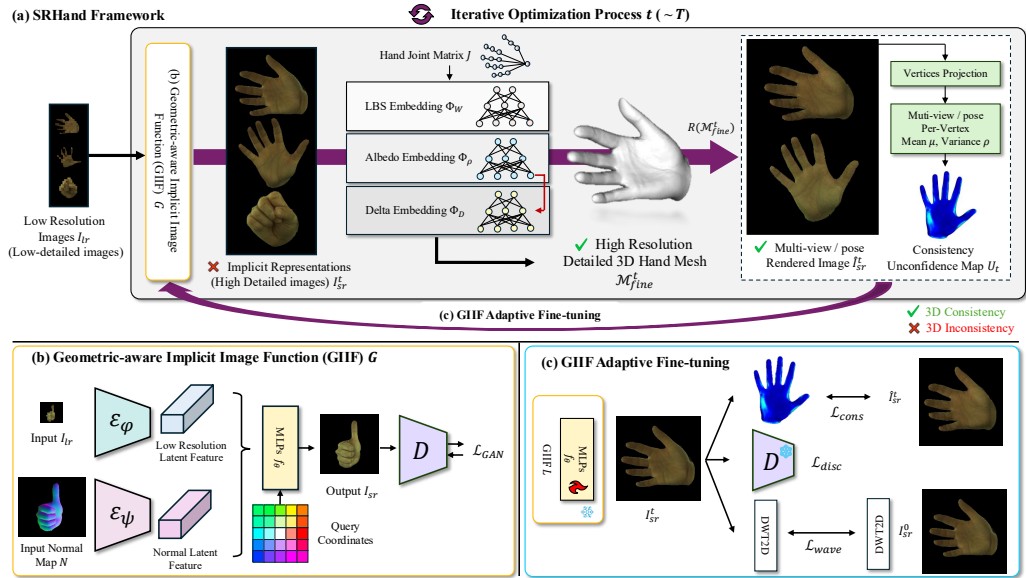

Figure 2: (a) Given LR images, we reconstruct high-resolution images using GIIF. Using these images, we reconstruct detailed 3D shapes while jointly optimizing the GIIF through adaptive fine-tuning from 3D shapes. (b) shows the architecture of GIIF. (c) represents the adaptive fine-tuning process.

geometric-aware implicit image function, Sec. 3.2 describes learning explicit hand representation, and Sec. 3.3 gives the details of 3D shape learning with adaptive fine tuning of the SR module.

## 3.1 Learning Implicit Image Function

**Preliminary.** The implicit image function interprets images as coordinate-based neural functions, enabling resolution-agnostic inference [6]. Its image function gets sampled encoded features using pixel coordinates $x$, and cell shape features $c$ to realize the query position of the overall shape cells. The implicit function is formulated as:

$$LIIF(I_{lr}) = f_\theta(z, [x, c]), \text{ where } z = E_\varphi(I_{lr}) \tag{1}$$

where $E_\varphi$ represents a low-resolution image encoder parameterized with $\varphi$, and $c = [c_h, c_w]$ contains two values specifying the height and width of a query pixel.

**Geometric-aware Implicit Image Function.** Shown in Fig. 2 bottom left, focusing on hand image SR, we propose a geometric-aware implicit image function (GIIF) that leverages surface normal guidance to recover high-frequency texture details via continuous image representations. Inspired by SuperNeRF-GAN [61], which integrates geometric priors for enhanced SR, GIIF utilizes a normal map derived from the MANO [45] template hand model. These normal maps provide structural priors for hand articulation, enabling robust reconstruction of fine-grained geometric details. Normal embedded features are then concatenated with LR image features, which are represented continuously through linear layers parameterized with $\theta$ with pixel coordinates queries.

Given a LR hand image $I_{lr}$, GIIF $G$ extracts hierarchical features using a Residual Dense Network (RDN) encoder $\mathcal{E}_\varphi$ (parameterized by $\varphi$) [60]. In parallel, the normal map $N$ extracted from the MANO [45] is encoded via a stacked hourglass encoder $\mathcal{E}_\psi$ (parameterized by $\psi$) [42] which is known effective for multi-scale feature extraction. The fused feature $\mathbf{f}_{fused}$ is obtained through:

$$\mathbf{f}_{fused} = \mathcal{E}_\varphi(I_{lr}). \oplus \mathcal{E}_\psi(N) \tag{2}$$

where $\oplus$ denotes channel-wise concatenation. Following LIIF [6], continuous coordinate queries $x$ are mapped to RGB values via linear layers $\mathcal{F}_\theta$ (parameterized by $\theta$):

$$I_{sr} = G(I_{lr}, N) = \mathcal{F}_\theta\left(\mathbf{f}_{fused}, [x, c]\right) \tag{3}$$

To enhance photorealism and sharper details, we adopt adversarial training loss $\mathcal{L}_{GAN}$ with a discriminator $\mathcal{D}$. The GIIF training loss functions are:

$$\mathcal{L}_{total} = \lambda_1 \mathcal{L}_1 + \lambda_{LPIPS} \mathcal{L}_{LPIPS} + \lambda_{GAN} \mathcal{L}_{GAN} \tag{4}$$

## 3.2 Learning Detailed Explicit Shapes

**Preliminary.** XHand [10] has achieved expressive hand avatar reconstruction from 2D high-resolution images. XHand subdivides MANO [45] mesh ($\bar{\mathcal{M}}$) 3 times, denoted as $\bar{\mathcal{M}}'$, increasing the resolution of the mesh. This approach enables the explicit shape to capture finer detailed geometric shapes. With subdivided hand meshes, XHand optimizes Linear Blend Skinning (LBS) weights $W$, delta values $D$ to calculate vertex displacements, and albedo colors $\rho$ from feature embedding modules $\Phi_{\{W,D,\rho\}}$ from given input hand joints $J$. XHand is formulated as:

$$\hat{I}_{\pi^i} = R(\pi^i|\mathcal{M}, J) = \Phi_\rho(J) \cdot SH(Y, \mathcal{N}'), \text{where } \mathcal{M} = \Omega(\bar{\mathcal{M}}' + \Phi_D(J), \Phi_W(J), \theta, \beta) \quad (5)$$

where $\theta$ and $\beta$ denote the pose and shape parameters for template mesh $\Omega$, respectively. $\pi^i$ is the camera parameter on $i$-th viewpoint, $\rho$ denotes albedo, $SH$ denotes spherical harmonics function with $Y$ as coefficients, and $\mathcal{N}'$ is the normals rendered from posed mesh $\mathcal{M}$.

**Enhancing Geometric Details on the Explicit Shape.** We adopt a high-resolution mesh representation that enables fine-grained geometric modeling of the hand surface. Our delta feature embedding module $\Phi_\rho$ incorporates the predicted color to disentangle geometric deformations from texture-driven appearance. To facilitate, we add the mean texture loss function to the training loss functions. Acquiring mean textures removes the appearance caused by texture color and encourages the model to learn geometric appearances. Our 3D representation is formulated as:

$$\mathcal{M}_{fine} = \Omega(\bar{\mathcal{M}}' + \Phi_D(J, \Phi_\rho(J)), \Phi_W(J), \theta, \beta) \quad (6)$$

## 3.3 Overall Pipeline.

Shown in Fig. 2, we first train GIIF to learn the implicit image priors of hand appearances through diverse hand subjects. After training GIIF, we get highly detailed images from given low-detailed images. However, similar to prior works [54, 57, 55, 26], GIIF itself does not guarantee 3D consistencies. To preserve 3D consistencies, we propose to fine-tune the SR model with 3D reconstructed models.

**Adaptive Fine-tuning.** Guo *et al.* [12] mathematically analyzed fine-tuning generative diffusion models with guided self-generated images to achieve global optima. We adaptively fine-tune GIIF with a reconstructed 3D shape trained on images generated by GIIF itself. As the 3D reconstruction model inherits consistencies across multi-view / pose diversities, iterative fine-tuning

---

**Algorithm 1** Psuedocode of SRHand fine-tuning.

**Require:** Dataset of $I_{lr}$, $J$ and $N$, pre-trained GIIF $G_{\psi,\varphi,\theta}$ and discriminator $D$, 3D reconstruction networks $\Phi$ involving subdivided template mesh $\mathcal{M}'$, $T$ total epochs, $T_{fine}$ total fine-tuning epochs, $t_{fine}$ interval steps for fine-tuning

**Ensure:** Personalized $\Phi_{W,D,\rho}$ and $G$
1: $I_{sr}^0 \leftarrow G(I_{lr}, N)$      ▷ Initially upscale $I_{lr}$ images
2: **for** $t = 0$ to $T - 1$ **do**
3:     $\{\hat{I}^t, \mathcal{M}_{fine}^t\} \leftarrow \Phi_{W,D,\rho}(J)$
4:     Compute $\mathcal{L}_{3D}(\hat{I}^t, \mathcal{M}_{fine}^t, I_{sr}^t)$ (Eq. 7)
5:     Gradient step to update $\Phi_{W,D,\rho}$
6:     **if** $t \bmod t_{fine} = 0$ **then**   ▷ Fine-tuning stage
7:         **for** $i = 0$ to $T_{fine} - 1$ **do**
8:             $I_{sr}^i \leftarrow L(I_{lr})$
9:             Compute $\mathcal{L}_{aft}(I_{sr}^i, I_{sr}^0, \hat{I}^t, \mathcal{M}_{fine}^t)$(Eq. 10)
10:            Gradient step to update $L_\theta$
11:         **end for**
12:         $I_{sr}^{t+1} \leftarrow L(I_{lr})$     ▷ Update the SR image
13:     **else**
14:         $I_{sr}^{t+1} \leftarrow I_{sr}^t$     ▷ If not, maintain $I_{sr}^t$
15:     **end if**
16: **end for**

---

GIIF shifts its distribution to the consistency maintained space. We show the pseudo code of our SRHand fine-tuning in Alg. 1 and the loss functions are introduced in Sec. 3.4.2.

## 3.4 Training Objectives

### 3.4.1 3D Geometric Loss.

Our loss function for learning 3D geometry includes a photometric loss $\mathcal{L}_{rgb}$ to minimize rendering errors via inverse rendering and a regression loss $\mathcal{L}_{reg}$ for hand geometry. $\mathcal{L}_{rgb}$ consists of L1 and perceptual loss between rendered image $\hat{I}$ and SR image $I_{sr}$. $\mathcal{L}_{reg}$ consists of part-aware Laplacian

smoothing $\mathcal{L}_{pLap}$ and edge regression loss $\mathcal{L}_{edge}$ used in [10]. $\mathcal{L}_{pLap}$ is $\sum_i \phi_{pLap}\mathbf{A}$ applied to both albedo $\rho$ and displacements $D$, where $\phi_{plap}$ is part-level weights and $\mathbf{A}$ is Laplace matrix. $\mathcal{L}_{edge}$ is $\sum_{i,j} \|\hat{e}_{ij} - e_{ij}\|_2^2$, where $e_{ij}$ is euclidean length between adjacent vertices $V_i$ and $V_j$ in the refining mesh $\mathcal{M}_{fine}$, and $\hat{e}_{ij}$ is the corresponding edge length in the initially subdivided mesh $\bar{\mathcal{M}}'$.

**Mean Texture Loss and Total Loss.**   To enhance details for geometric appearance, we apply the mean texture loss function $\mathcal{L}_{mt}$ utilizing the perceptual loss to capture fine details. $\mathcal{L}_{3D}$ is defined as:

$$\mathcal{L}_{3D} = \mathcal{L}_{rgb} + \mathcal{L}_{reg} + \lambda_{mt} \underbrace{\mathcal{L}_{LPIPS}(R(\pi|\bar{\Phi}_\rho^t), I_{sr}^t)}_{\mathcal{L}_{mt}} \tag{7}$$

where $\bar{\Phi}_\rho^t$ represents the mean texture of albedo colors.

### 3.4.2   Adaptive Fine-tuning Loss.

Our adaptive fine-tuning loss function consists with three parts; the consistency loss $\mathcal{L}_{cons}$, frequency maintenance loss $\mathcal{L}_{wave}$, and discriminator loss $\mathcal{L}_{disc}$.

**Consistency Loss.**   The consistency loss aims to minimize the variation of the vertex queries $Q \in \mathbb{R}^{N \times 3}$ between multi-view and multi-pose images $I_{sr}$. We first calculate the mean $\mu_t$ and variance $\rho_t$ of the given each queries from multi-view images at sequence $t$. After earning $\mu_t, \rho_t$ for all $Q$, we create an unconfidence map $U^t(Q)$. Pose consistencies are indicated as subtraction of $\mu_t$ with previous step $\mu_{t-1}$ and multi-view consistencies are presented as $\rho_t$ variances. Having $U^t(Q)$ as $(\mu_t - \mu_{t-1}) + \rho_v$, we formulate $\mathcal{L}_{cons}$ as:

$$\mathcal{L}_{cons} = \mathcal{L}_1((R(\mathcal{M}_{fine}) \cdot U^t(Q), I_{sr} \cdot U^t(Q)) \tag{8}$$

**Frequency Maintenance Loss.**   Applying $\mathcal{L}_{cons}$ leads to the mean texture value losing high-frequency details. To maintain total high-frequency details, we adopt $\mathcal{L}_{wave}$ as:

$$\mathcal{L}_{wave} = \mathcal{L}_1(\tilde{\phi}(I_{sr}^t), \tilde{\phi}(I_{sr}^0))) + \mathcal{L}_1(\sum \widehat{\phi}(I_{sr}^t), \sum \widehat{\phi}(I_{sr}^0)) \tag{9}$$

where $\tilde{\phi}$ and $\widehat{\phi}$ denote discrete wavelet transform functions with low-pass and high-pass filters.

**Discriminator and Total Loss.**   To improve photometric reality, we propose to use the discriminator $D$ trained with GIIF. We include the discriminator loss $\mathcal{L}_{disc}$ to the total adaptive fine-tuning loss function $\mathcal{L}_{aft}$, which is described as follows:

$$\mathcal{L}_{aft} = \mathcal{L}_{cons} + \mathcal{L}_{wave} + \mathcal{L}_{disc} \text{ where } \mathcal{L}_{disc} = log(1 - D(I_{sr}^t)) \tag{10}$$

## 4   Experiments

### 4.1   Datasets and Experiment Settings

**InterHand2.6M [40].**   We mainly use the InterHand2.6M [40] dataset for our super-resolution and 3D reconstruction experiments. The dataset presents diverse views of cameras with a single hand and

Table 1: Quantitative comparisons between compared methods using InterHand2.6M [40] and Goliath [35] dataset. PSNR / LPIPS (SR) shows the PSNR and LPIPS performance of the super-resolution modules. Mark "Incon." stands for inconsistency and "ftd." stands for the fine-tuned model. The top three results are highlighted in red, orange, and yellow, respectively.

| SR Module | 3D Recon. Methods | InterHand2.6M [40] | | | | Goliath [35] | | | |
|---|---|---|---|---|---|---|---|---|---|
| | | PSNR / LPIPS (SR) | PSNR | LPIPS | P2P (mm) | Incon. | PSNR / LPIPS (SR) | PSNR | LPIPS | P2P (mm) | Incon. |
| Bicubic | Ours | 22.23 / 0.2645 | 26.44 | 0.0895 | 4.01 | 0.0131 | 19.17 / 0.3244 | 22.52 | 0.1377 | 5.80 | 0.0128 |
| LIIF [6] | XHand [10] | 27.47 / 0.1063 | 27.36 | 0.0691 | 4.32 | 0.0151 | 24.87 / 0.1459 | 23.64 | 0.0984 | 3.34 | 0.0146 |
| | Ours | | 27.11 | 0.0755 | 3.39 | 0.0151 | | 22.87 | 0.1123 | 4.12 | 0.0140 |
| GIIF (w/o ftd.) | UHM [39] | 29.96 / 0.0305 | 22.33 | 0.1522 | 72.55 | - | 27.91 / 0.0497 | 23.85 | 0.1319 | 24.29 | - |
| | XHand | | 27.71 | 0.0507 | 3.43 | 0.0067 | | 22.76 | 0.1118 | 3.70 | 0.0084 |
| | Ours | | 29.17 | 0.0404 | 3.09 | 0.0058 | | 23.50 | 0.0783 | 3.49 | 0.0070 |
| GIIF (w/ ftd.) | XHand | 30.03 / 0.0303 | 28.75 | 0.0443 | 3.45 | 0.0052 | 28.07 / 0.0495 | 21.95 | 0.1139 | 3.60 | 0.0082 |
| | Ours | 30.06 / 0.0302 | 29.88 | 0.0362 | 2.16 | 0.0050 | 28.09 / 0.0495 | 24.31 | 0.0813 | 3.50 | 0.0069 |

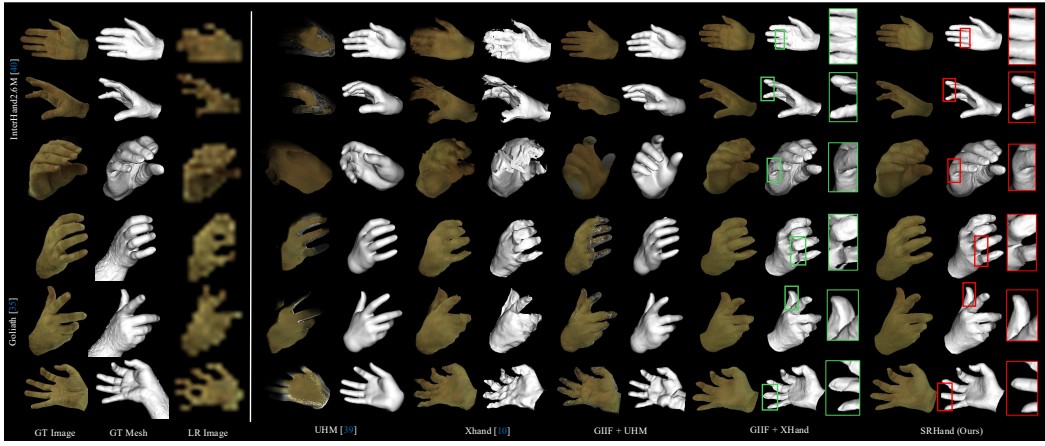

Figure 3: Qualitative results of 3D reconstructions. Given low-resolution (low-detailed) images, SRHand reconstructs high-resolution personalized hand meshes.

two hands of different identities. For SR experiments, we divide train and validation data based on capture subjects to guarantee robustness across subjects in the "train" split. For 3D reconstruction experiments, we used *Capture0* and *Capture1* in the "test" split following prior works [5, 41, 13, 10]. We reconstruct 3D hands with 20 views and 20 frames and evaluate with the remaining frames.

**Goliath [35].**    Goliath [35] dataset has similar experiment settings with InterHand2.6M [40], however, it presents scanned mesh and high-resolution images. However, as it only presents only 4 subjects, which is insufficient for training the SR module, we used the SR module trained on Inter-Hand2.6M. More details on preprocessing the Goliath dataset are presented in the supplementary.

**Metrics.**    For the quantitative comparisons of different appearance models, we use a set of metrics that are often applied to assess the fidelity and quality of rendered images. We mainly use the learned perceptual image patch similarity (LPIPS) [59], the structural similarity metric (SSIM) [51], and the peak signal-to-noise ratio (PSNR) for comparison metrics. We also present Point-to-Point (P2P) of reconstructed meshes against the ground truth meshes. Since InterHand2.6M [40] does not provide scanned meshes, we trained XHand [10] on 50 frames from 50 viewpoints using high-resolution ground truth images. The obtained meshes serve as our pseudo ground truth for P2P evaluation.

## 4.2    Compared Methods

We conduct experiments in three parts to investigate the efficiency of our proposed framework: (1) 3D Reconstruction from SR Images, (2) 3D Consistency Considered Super-Resolution, and (3) Hand Image Super-Resolution.

**3D Reconstruction from SR Images.**    Our method leverages 3D appearances for both 3D representation and SR module fine-tuning. As shown in Tab. 1, we evaluate 3D reconstruction performance and fine-tuned SR model across different alternative methods including UHM [39]. For both InterHand2.6M [40] and Goliath [35] datasets, 3D reconstruction results are reported in terms of PSNR, LPIPS for visual appearance, and point-to-point (P2P) distance for geometric shapes. The inconsistency is quantified using aggregated variations across reconstructed vertices.

Fig. 3 and Tab. 1 show the experimental results on the InterHand2.6M [40] and Goliath [35] datasets. Our hand mesh better represents hand shapes than the compared methods. Specifically, UHM [39] fails to represent hand shapes and textures, losing details and including background artifacts. Simply attaching GIIF and XHand [10] results in overbounded shapes due to inconsistencies of hand shape and details in the SR images. Furthermore, the results demonstrate that fine-tuning the SR module enhances its 3D reconstruction performance compared to its original state. This improvement highlights the effectiveness of adaptive fine-tuning in enforcing consistency for more accurate 3D reconstruction while improving photometric quality, as shown by the PSNR/LPIPS (SR) metric. In conclusion, SRHand achieves the best overall performance, outperforming all tested configurations.

Table 2: Quantitative comparisons among 3D consistency considered SR methods. (×) denotes an upscaling factor. Bold and underline denote top two scores.

| Upscale factor | Methods | PSNR | LPIPS | SSIM |
|---|---|---|---|---|
| ×4 | NeRF-SR [47] | 28.83 | 0.1410 | 0.8638 |
| | DiSR-NeRF [26] | 28.82 | 0.1303 | 0.8736 |
| | SRGS [9] | **32.74** | 0.0731 | **0.9335** |
| | SRHand | 29.06 | **0.0475** | 0.9016 |
| ×8 | SRGS | **29.22** | 0.1520 | 0.8739 |
| | SRHand | 28.86 | **0.0487** | **0.9007** |
| ×16 | SRGS | 27.18 | 0.3872 | 0.7899 |
| | SRHand | **27.52** | **0.0566** | **0.8858** |

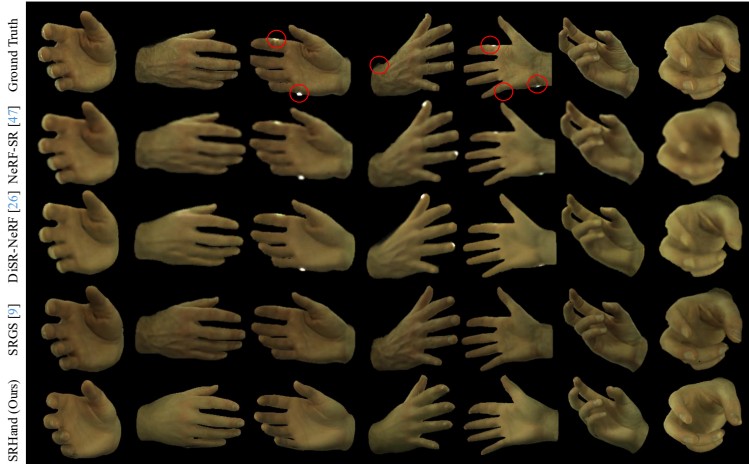

Figure 4: Qualitative results. Red circles show background artifacts in GT data. All baselines are compared with upscaling factor 4.

**3D Consistency Considered Super-Resolution.** We compare SRHand with NeRF-SR [47], DiSR-NeRF [26] and SRGS [9] for 3D consistency considered super-resolution. Prior works are restricted to reconstructing static objects rather than human avatars, thus we train our method with only a single pose assuming it as a static scene with multi-view images for fair comparison. In addition, prior works are also restricted to × 4 scale upsampling, we present our results with × 4 and × 16 upsampling results. SuperNeRF [57] and Yoon *et al.* [55] are omitted due to the unavailability of source codes and lack of implementation details. However, note DiSR-NeRF shows better performance than SuperNeRF, as reported in [26].

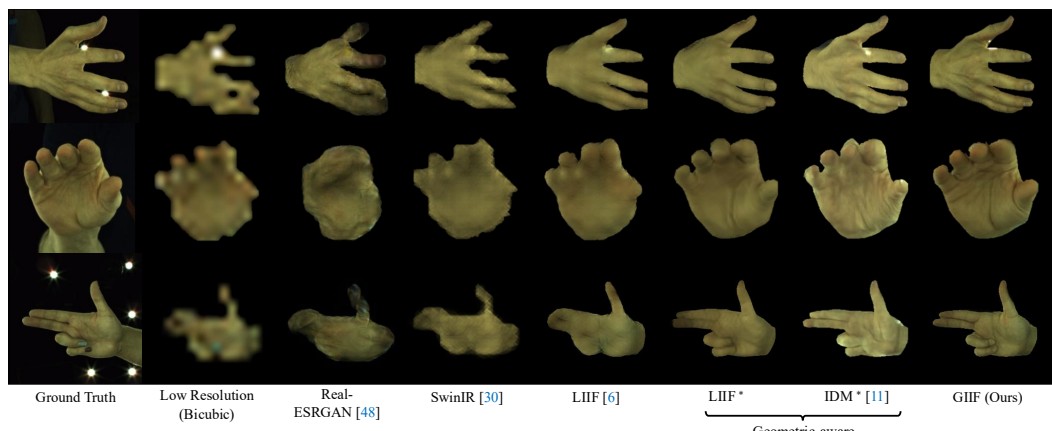

Figure 5: Qualitative comparisons among SR modules. Original IDM results are in the supplementary.

Table 3: Quantitative comparisons of SR modules on InterHand2.6M [40].

(a) Experiments are performed with ×16 upscaling. (* denotes the model has been modified with normal map conditioning.)

| Methods | | PSNR ↑ | LPIPS ↓ |
|---|---|---|---|
| Real-ESRGAN [48] | | 22.39 | 0.2287 |
| SwinIR [30] | | 25.51 | 0.1552 |
| LIIF [6] | | 25.76 | 0.1848 |
| IDM [11] | | 14.58 | 0.3603 |
| Geometric-aware | LIIF* | 29.85 | 0.0996 |
| | IDM* | 21.49 | 0.0970 |
| | GIIF | **31.60** | **0.0637** |

(b) Results in (PSNR / LPIPS) of continuous scale trained on × 16 factor. GIIF achieves best performance in all scaling factors.

| Methods | Upscaling factor | | |
|---|---|---|---|
| | ×8 | ×21.3 | ×32 |
| LIIF [6] | 28.62/0.1319 | 23.91/0.2071 | 21.46/0.2693 |
| IDM [11] | 25.46/0.1215 | 21.28/0.1796 | 19.02/0.2426 |
| LIIF* | 30.20/0.0812 | 29.17/0.0855 | 28.55/0.0885 |
| IDM* | 22.68/0.0696 | 22.71/0.0780 | 22.87/0.0878 |
| GIIF | **32.70/0.0533** | **31.53/ 0.0606** | **30.01/0.0640** |

Tab. 2 shows quantitative results compared with NeRF-SR [47], DiSR-NeRF [26] and SRGS [9] trained on 40 view images. SRHand supports arbitrary-scale super-resolution and outperforms prior methods, even at a 16 times upscaling factor. Fig. 4 shows the qualitative results and illustrates that our method captures fine hand features compared to prior works. NeRF-SR and DiSR-NeRF still suffer from blurriness and appear overly synthetic. Considering that GT data and previous works often include background colors on the hand, our method eliminates these artifacts. Despite having differences from GT data, our approach still achieves superior performance.

**Hand Image Super-Resolution.** We compare GIIF with existing image super-resolution models [48, 30, 6, 11]. We retrained all the baselines to the hand image datasets and extend LIIF [6] to geometric-aware LIIF (denoted as LIIF*) and IDM [11] to geometric-aware IDM (denoted as IDM*) for a fair comparison. Tab. 3a shows the quantitative comparisons and Fig. 5 shows the qualitative results. GIIF significantly achieves better results with more realistic hands with detailed appearances. Comparing LIIF with LIIF* and IDM and IDM*, we observe that the normal conditioning substantially improves rendering quality, enabling part-aware generation. Furthermore, the discriminator refines photometric details and realism. Tab. 3b shows that our method achieves the highest performance at arbitrary upscaling factors, showing robustness across in-distribution and out-of-distribution scales.

Table 4: Ablation study of geometric enhancements and adaptive fine-tuning loss functions.

(a) Ablation study on 3D hand reconstruction methods using two identities.

| | | PSNR | LPIPS | SSIM |
|---|---|---|---|---|
| Identity 1 | XHand [10] | 27.77 | 0.0423 | 0.8977 |
| | Ours | **28.92** | **0.0353** | **0.9089** |
| Identity 2 | XHand | 28.72 | 0.0338 | 0.9125 |
| | Ours | **29.42** | **0.0303** | **0.9189** |

(b) Ablation study on the adaptive fine-tuning loss functions.

| $L_{cons}$ | $L_{wave}$ | $L_{disc}$ | PSNR (SR) | PSNR | LPIPS | P2P ($mm$) |
|---|---|---|---|---|---|---|
| | | | 29.96 | 29.17 | 0.0404 | 3.09 |
| ✓ | | | 23.94 | 23.39 | 0.0864 | 3.72 |
| ✓ | ✓ | | 29.93 | 28.97 | 0.0410 | 2.41 |
| ✓ | ✓ | ✓ | **30.06** | **29.88** | **0.0362** | **2.16** |

## 4.3 Ablation Studies

**Geometric Enhancements.** We conduct an ablation study to compare our enhanced mesh representation with the baseline XHand [10]. As shown in Tab. 4a and Fig. 6c, our representation better captures fine details while consistently achieving higher performance across all metrics in different identities.

**Effectiveness of Adaptive Fine-tuning Loss.** We show the ablation study of adaptive fine-tuning loss. Tab. 4b and Fig. 6a represent that mesh distance error and texture inconsistencies decrease when adaptive fine-tuning is applied.
• *Consistency Loss.* We evaluate the impact of various loss components during fine-tuning, shown in Fig. 6b, and Tab. 4b. Applying the consistency loss ($L_{cons}$) degrades the performance leading overall hand textures to a mean value. However, regarding Fig. 6, consistencies are well maintained presenting the mean texture.

Table 5: Quantitative results of showing the effectiveness of GT parameters.

| | MPJPE (*mm*) | Super-Resolution | | 3D Reconstruction | |
|---|---|---|---|---|---|
| | | PSNR | LPIPS | PSNR | LPIPS |
| Non-using GT Param. | 9.54 | **30.11** | 0.0306 | 27.18 | 0.0635 |
| Using GT Param. | - | 30.06 | **0.0302** | **29.88** | **0.0362** |

• *Frequency Maintenance Loss.* The wavelet loss ($L_{wave}$) plays a vital role in preserving multi-scale details in the upsampling and reconstruction process. As shown in Tab. 4b and Fig. 6b, its inclusion alongside the consistency loss significantly enhances visual quality by maintaining high-frequency components. This demonstrates that the frequency maintenance loss is effective for maintaining structural details and fine textures.

• *Discriminator Loss.* Adding the discriminator loss ($L_{disc}$) further refines the output by promoting photometric consistency and realism. When integrated with $L_{cons}$ and $L_{wave}$, $L_{disc}$ improves both quantitative and qualitative metrics. This adversarial component enhances the generation of realistic textures, as shown in Tab. 4b and Fig. 6b.

**Non-using GT parameters.** Although using ground-truth (GT) template parameters is a widely adopted strategy in 3D hand and body avatar reconstruction [5, 20, 29, 10, 62, 37], and our method follows the same strategy, we additionally provide results where noise is added to the GT parameters. Tab. 5 shows that MANO parameter errors do not have a critical effect on super-resolution performance, while they affect the quality of 3D hand reconstruction.

## 5   Conclusions

In this paper, we present SRHand that integrates view/pose-aware implicit neural representations with explicit 3D mesh reconstruction for high-fidelity hand avatar modeling. Our approach leverages a geometric-aware implicit image function (GIIF) to super-resolve low-detailed hand images in arbitrary scale while maintaining 3D view/pose consistency through jointly fine-tuning the SR module and 3D reconstruction. Extensive quantitative and qualitative results using InterHand2.6M [40] and Goliath [35] datasets demonstrate that our method efficiently captures fine details in both 2D images and 3D shapes of hands, and outperforms the baseline methods. We anticipate that our approach can be applied to a wider scope of avatar creation, providing users with a realistic experience in virtual environments and accurate 3D human reconstruction.

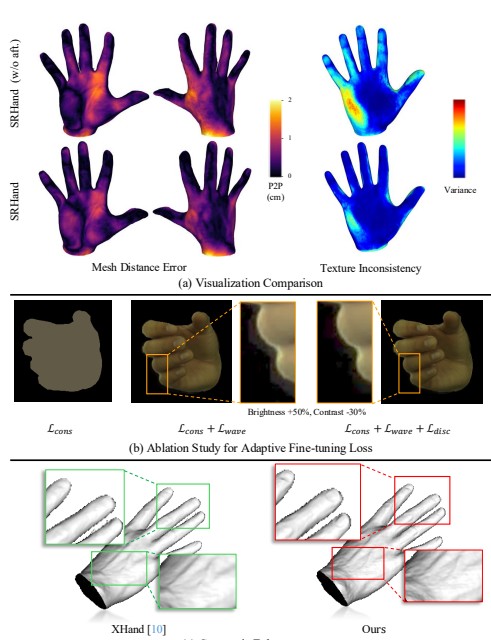

Figure 6: Visualization of ablations studies: (a) Inconsistencies and mesh distance error. Mark "aft." stands for adaptive fine-tuning. (b) $\mathcal{L}_{aft}$ loss function. To better highlight the difference, we adjust brightness and contrast. (c) Enhancing geometric details.

## 6   Acknowledgement

This work was supported by NST grant (CRC 21011, MSIT), IITP grant (RS-2023-00228996, RS-2024-00459749, RS-2025-25443318, RS-2025-25441313, MSIT) and KOCCA grant (RS-2024-00442308, MCST).

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
