# OpenReview forum: "SRHand: Super-Resolving Hand Images and 3D Shapes via View/Pose-aware Neural Image Representations and Explicit Meshes"
_NeurIPS.cc/2025/Conference — NeurIPS 2025 poster_

### Official Review · Reviewer_hSnG · 2025-06-16

**Clarity:** 2
**Significance:** 3
**Originality:** 2
**Rating:** 5
**Confidence:** 5

**Summary:**

This paper proposes SRHand, a novel framework for super-resolving low-resolution hand images and reconstructing high-fidelity 3D hand meshes. The method integrates a Geometric-aware Implicit Image Function (GIIF) for continuous image upsampling with an explicit high-resolution mesh-based 3D reconstruction module. Furthermore, an adaptive fine-tuning strategy is introduced to enforce multi-view and pose consistency by jointly optimizing the image super-resolution module and explicit 3D shape. Experiments on InterHand2.6M and Goliath datasets show SOTA performance both quantitatively and qualitatively, demonstrating the effectiveness in reconstructing highly detailed hand avatars from low-resolution images.

**Questions:**

1. Define variant “NAIIF” in Figure 2 and Figure 4.
2. Please explain the innovation about GIIF module.

**Ethical Concerns:**

["NO or VERY MINOR ethics concerns only"]

**Final Justification:**

According to the authors’ rebuttal and additional experiments, I increase my score. Thanks for the additional explanation. I hope the authors can add these content in the final version.

**Limitations:**

- Ambiguity in Terminology. The paper introduces GIIF as the main SR module. However, Figure 2 and Figure 4 labels one variant as “NAIIF”, which is not formally defined in the main text.
- The GIIF module mainly reuses known components: normal maps as structural priors (as used in SuperNeRF-GAN [2]), and adversarial losses (as in LIIF-GAN [1]). In particular, SuperNeRF-GAN proposed to use the normal maps as geometric priors to guide the super-resolution process within a NeRF-based framework. While the integration is novel in the context of hand avatars, the module itself lacks substantial architectural innovation.

[1] Yinbo Chen, Sifei Liu, and Xiaolong Wang. Learning continuous image representation with local implicit image function. In CVPR, 2021.

[2] Peng Zheng, Linzhi Huang, Yizhou Yu, Yi Chang, Yilin Wang, and Rui Ma. Supernerf-gan: A universal 3d-consistent super-resolution framework for efficient and enhanced 3d-aware image synthesis. arXiv.2501.06770, 2025.

**Quality:**

3

**Strengths And Weaknesses:**

- The paper tackles the under-explored but practically important problem of reconstructing detailed hand geometry from low-resolution multi-view inputs, which is common in full-body capture pipelines.
- The GIIF module is technically well-motivated. The use of normal maps as 3D geometric priors has been validated in several human reconstruction works, such as ECON [3], which demonstrates that normals can effectively guide surface detail recovery. Building upon this idea, the GIIF leverages normal maps from the MANO to introduce structural awareness into the image super-resolution process.
- The proposed method significantly outperforms baseline SR and 3D reconstruction methods in PSNR, LPIPS, SSIM, and P2P metrics. Ablation studies further support the effectiveness of each component.

[1] Yuliang Xiu, Jinlong Yang, Dimitrios Tzionas, and Michael J. Black. ECON: Explicit Clothed Humans Optimized via Normal Integration. In CVPR, 2023.

---

> ### Author Rebuttal · Authors · 2025-07-25
>
> ## Common Response to Reviewers
> We sincerely thank the reviewers for their constructive feedback. We appreciate all reviewers recognizing the motivation of our work and the novelty of the method, along with the SOTA results. We have conducted several additional experiments to address remaining concerns.
>
> 1. 3D reconstruction experiment with other hand reconstruction methods: HARP[R1] , LiveHand [R2].
>
> 2. 3D Consistency-considered super-resolution experiment including Gaussian splatting, SRGS [R3].
>
> 3. Extreme resolution experiment to show the lower-bound of SRHand, and the results for different occlusion rates.
>
> Please refer to the additional experiment results for your consideration.
>
> References:
> [R1] Korrawe Karunratanakul, Sergey Prokudin, Otmar Hilliges, and Siyu Tang. Harp: Personalized hand reconstruction from a monocular rgb video. In CVPR, 2023.
>
> [R2] Akshay Mundra, Mallikarjun B R, Jiayi Wang, Marc Habermann, Christian Theobalt, and Mohamed Elgharib. Livehand: Real-time and photorealistic neural hand rendering. In ICCV, 2023.
>
> [R3] Xiang Feng, Yongbo He, Yan Yang, and Yubo Wang, Yifei Chen, Zhan Wang, Zhenzhong Kuang, Feiwei Qin, Jiajun Ding. SRGS: Super-Resolution 3D Gaussian Splatting. arXiv:2404.10318, 2024.
>
>
> **Q1: Ambiguity in Terminology.**
> A1: We had included ***'known_typo_main_figure.pdf**'* in the submitted supplementary file mentioning those typos.
>
> **Q2: Innovation about GIIF module.**
> A2: The main contribution of SRHand lies not in GIIF alone, but in the joint optimization of the SR module and the 3D reconstruction module, which enables consistent super-resolved images and high-resolution animatable hand meshes to be learned simultaneously. GIIF supports this pipeline as GIIF key modules are used in the joint optimization and affect the 3D reconstruction stage. Integrating GIIF into a tightly coupled SR–3D pipeline for dynamic, low-resolution hand avatars, to the best of our knowledge, is a novel contribution. This differs fundamentally from prior image SR methods, which operate independently of 3D learning or induced by 3D consistency for static objects or scenes.
>
> Replacing GIIF with LIIF (or LIIF* variants) leads to a noticeable drop in both SR quality (Tab. 1) and 3D reconstruction accuracy (Tab. 3). Importantly, the adaptive fine-tuning stage, where GIIF and the mesh module are updated together, reduces P2P error from 3.09*mm* to 2.16*mm*, highlighting that our novelty stems from the SR–3D joint optimization and module sharing across stages.

---

> ### Author Response · Authors · 2025-08-06
>
> Thank you again for your constructive feedback. We have provided a rebuttal to address your concerns, and we hope that our answers have sufficiently addressed them. If there are any remaining points you would like us to clarify, we would greatly appreciate your further feedback and would be glad to discuss them.

---

### Official Review · Reviewer_sv5s · 2025-06-23

**Clarity:** 3
**Significance:** 3
**Originality:** 3
**Rating:** 5
**Confidence:** 4

**Summary:**

The paper introduces the SRHand framework, which combines geometry - aware implicit image functions with explicit 3D mesh optimization to reconstruct detailed 2D hand images and 3D shapes from low - resolution images. It shows superior performance on InterHand2.6M and Goliath datasets compared to existing methods.

**Questions:**

pls refer to weaknesses.

**Ethical Concerns:**

["NO or VERY MINOR ethics concerns only"]

**Final Justification:**

Thank you for your rebuttal—it has resolved all my concerns. I will raise my score to 5 and hope the authors will revise the paper in future versions according to the rebuttal.

**Limitations:**

yes

**Quality:**

4

**Strengths And Weaknesses:**

### **Strengths**

(1) Novelty: It's the first to merge implicit neural representations (GIIF) with explicit reconstruction mesh, ensuring multi - view/pose consistency via normal - map priors and reducing reliance on high - resolution inputs.

(2) Comprehensive experiments and strong detail reconstruction: It achieves a PSNR of 31.60 under x16 super - resolution, captures fine - grained features like wrinkles and nails.

(3) Open - source code and high reproducibility.

(4) SRHand has great potential for practical deployment and engineering implementation.

### **Weaknesses**

(1) Can it show detail - reconstruction results under extremely low - resolution inputs and test the lower limit?

(2) Is there validation under challenging conditions like complex occlusions?

(3) Missing related references, such as I2UV - HandNet (iccv 2021), which performs super - resolution of 3D hand meshes based on UV maps.

---

> ### Author Rebuttal · Authors · 2025-07-31
>
> ## Common Response to Reviewers
> We sincerely thank the reviewers for their constructive feedback. We appreciate all reviewers recognizing the motivation of our work and the novelty of the method, along with the SOTA results. We have conducted several additional experiments to address remaining concerns.
>
> 1. 3D reconstruction experiment with other hand reconstruction methods: HARP[R1] , LiveHand [R2].
>
> 2. 3D Consistency-considered super-resolution experiment including Gaussian splatting, SRGS [R3].
>
> 3. Extreme resolution experiment to show the lower-bound of SRHand, and the results for different occlusion rates.
>
> Please refer to the additional experiment results for your consideration.
>
> References:
> [R1] Korrawe Karunratanakul, Sergey Prokudin, Otmar Hilliges, and Siyu Tang. Harp: Personalized hand reconstruction from a monocular rgb video. In CVPR, 2023.
>
> [R2] Akshay Mundra, Mallikarjun B R, Jiayi Wang, Marc Habermann, Christian Theobalt, and Mohamed Elgharib. Livehand: Real-time and photorealistic neural hand rendering. In ICCV, 2023.
>
> [R3] Xiang Feng, Yongbo He, Yan Yang, and Yubo Wang, Yifei Chen, Zhan Wang, Zhenzhong Kuang, Feiwei Qin, Jiajun Ding. SRGS: Super-Resolution 3D Gaussian Splatting. arXiv:2404.10318, 2024.
>
> **Q1: Experiments on extremely small image resolution and complex occlusions.**
> A1: Our main experiments, ×16 upscale factor, already target a realistic setting in both human and hand reconstruction literature, where the hand occupies 1.5% (L24) of the full body size. To further probe the lower limit of SRHand, we conducted an additional experiment with an upscaling factor of ×32 (where the input images are as small as 8×8). While this factor goes beyond typical practical scenarios, it serves as a stress test to evaluate how well the joint SR–3D pipeline preserves geometry and texture. Thanks to the strong combination of the SR module and the 3D reconstruction module, the joint optimization maintains comparable performance even under this extreme setting, as shown in the table below. Furthermore, seeing the arbitrary input resolution experiments presented in Supplementary Section 3.5 together, these results demonstrate that SRHand scales consistently: as the input image resolution increases, 3D reconstruction quality improves accordingly.
>
>
> **Table 1:** Quantitative results of the arbitrary scale input image experiment, including an extreme resolution image.
> | Upscale factor | Super-Resolution PSNR | Super-Resolution LPIPS | 3D Reconstruction PSNR | 3D Reconstruction LPIPS |
> |----------------|-----------------------|------------------------|------------------------|-------------------------|
> | ×32            | 28.82                 | 0.0341                 | 28.23                  | 0.0409                  |
> | ×16            | 30.06                 | 0.0302                 | 29.88                  | 0.0362                  |
> | ×10.6          | 30.52                 | 0.0293                 | 30.08                  | 0.0343                  |
>
>
> **Q2: Validation under complex occlusions.**
> A2: We also present complex occlusion experiments. Hand reconstruction is well-known to be challenging due to frequent self-occlusions and highly articulated poses. Our multi-view video setup is designed to alleviate these issues, yet detailed geometric reconstruction remains difficult in occluded regions. To simulate such challenging conditions, we reduce the number of training cameras to one or two and sample pose-varying frames. The occlusion rate is quantified through the percentage of invisible vertices of the hand in the camera viewing space. As expected, the reconstruction accuracy decreases as the occlusion rate increases, shown in the table below; however, not drastically.
>
> **Table 2:** Quantitative results under diverse occlusion rate.
> | Occlusion Rate (Train View #) | PSNR  | LPIPS  | SSIM   |
> |------------------------------|-------|--------|--------|
> | 24.7% (1)                    | 25.54 | 0.0704 | 0.8636 |
> | 19.8% (1)                    | 26.56 | 0.0597 | 0.8874 |
> | 9.43% (2)                    | 28.05 | 0.0510 | 0.8636 |
> | 0%                           | 29.88 | 0.0362 | 0.9206 |
>
>
> **Q3: Missing Reference.**
> A3: I2UV-HandNet [1] focuses on reconstructing a high-resolution UV-map hand mesh (without texture) from a single image, which has a different objective compared to ours. Nevertheless, its combination of super-resolution and hand geometry reconstruction shares conceptual similarities with our approach, and we will include I2UV-HandNet as a related work in the final version.
>
>
> Reference:
> [1] Ping Chen, Yujin Chen, Dong Yang, Fangyin Wu, Qin Li, Qingpei Xia, Yong Tan. I2UV-HandNet: Image-to-UV Prediction Network for Accurate and High-fidelity 3D Hand Mesh Modeling. In ICCV, 2021.

---

> > ### Comment · Reviewer_sv5s · 2025-08-03
> > **Response**
> >
> > Thank you for your rebuttal—it has resolved all my concerns. I will raise my score to 5 and hope the authors will revise the paper in future versions according to the rebuttal.

---

### Official Review · Reviewer_dx64 · 2025-06-29

**Clarity:** 3
**Significance:** 2
**Originality:** 3
**Rating:** 4
**Confidence:** 3

**Summary:**

The paper describes a framework for super-resolving hand avatars from very low resolution multi-view images. It integrates an image SR and a 3D hand reconstruction module to model high fidelity avatars. It uses an implicit image function that is conditioned on surface normals extracted from a MANO model, making it geometry aware - called Geometric-aware Implicit Image Function (GIIF).
The method is validated on real-world datasets, demonstrating improvements in appearance and geometry quality, such as wrinkles, nails, and shape variations.

It uses a high detailed mesh like XHand, but does a better job in disentangling the color changes due to texture vs. geometry. This is done by combining LIIF-GAN style adversarial implicit function learning (to support any resolution) with MANO based normals for geometry awareness, followed by fine tuning. On InterHand and Goliath, reported PSNR, LPIPS, and P2P metrics show very significant improvements over competitors.

Significance is a bit unclear as the paper doesn't discuss where it's getting the pose estimates, and how robust the method is to imperfect pose. With lower resolution, hand pose also degrades, but the paper seems to ignore it. The scenario where the method is applied seems very unrealistic otherwise, where we have images that barely look like hands, but perfect hand pose somehow.

**Questions:**

My main question is around what happens with bad or noisy initial poses. If the system is not robust at all the noisy J, it shouldn't be attempting to show results with super low resolution input, as it makes no sense. Ideally the pose should also come from the same type of LR images if this is expected to have any real world impact.

I may have misunderstood the paper of course, in which case I'd be happy to increase my score.

**Ethical Concerns:**

["NO or VERY MINOR ethics concerns only"]

**Final Justification:**

I'd like to thank the authors for the discussion. After seeing the results without the GT pose, I decided to increase my score to borderline accept.

**Limitations:**

Basically all my questions about robustness to pose noise apply here.

**Quality:**

3

**Strengths And Weaknesses:**

Strengths:
- novel integration of a LIIF-GAN with explicit 3D meshes, combining the advantages of continuous, arbitrary-resolution detail generation from implicit representations with the structured geometric prior and animatability provided by explicit meshes.

- fine-tuning mechanism enforces multi-view and pose consistency for highly deformable hands.

- extensive quantitative and qualitative results show clear improvement over the SOTA.

Weaknesses:
- The paper seems to assume that hand pose is coming from somewhere, and it doesn't discuss what happens when the initial pose is noisy or simply bad. The input images used in results look ridiculously distorted such that no hand tracker would be able to extract any meaningful pose from them, so I think it's clear that the final pose completely depends on having a perfect pose estimate. The fine tuning stage doesn't clearly explain if pose itself is refined, but it doesn't look like it. In any case ablation study or limitations section don't mention what happens with bad poses either. My feeling is that if the paper assumes it will have perfect GT pose from 34x34 hand images somehow, it has less significance than it claims, because it can't be done. If it can actually refine the very bad initial poses, and get results that are so close to GT, then it has even more significance then it claims in the paper, but doesn't mention it.

- The paper does not directly compare against SOTA that leverage 3D Gaussian Splatting or diffusion models for 3D content generation. E.g. "3DEnhancer: Consistent Multi-View Diffusion for 3D Enhancement" by Luo et al, which looks like it could also do a good job with hands with some adaptation.

---

> ### Author Rebuttal · Authors · 2025-07-31
>
> ## Common Response to Reviewers
> We sincerely thank the reviewers for their constructive feedback. We appreciate all reviewers recognizing the motivation of our work and the novelty of the method, along with the SOTA results. We have conducted several additional experiments to address remaining concerns.
>
> 1. 3D reconstruction experiment with other hand reconstruction methods: HARP[R1] , LiveHand [R2].
>
> 2. 3D Consistency-considered super-resolution experiment including Gaussian splatting, SRGS [R3].
>
> 3. Extreme resolution experiment to show the lower-bound of SRHand, and the results for different occlusion rates.
>
> Please refer to the additional experiment results for your consideration.
>
> References:
> [R1] Korrawe Karunratanakul, Sergey Prokudin, Otmar Hilliges, and Siyu Tang. Harp: Personalized hand reconstruction from a monocular rgb video. In CVPR, 2023.
>
> [R2] Akshay Mundra, Mallikarjun B R, Jiayi Wang, Marc Habermann, Christian Theobalt, and Mohamed Elgharib. Livehand: Real-time and photorealistic neural hand rendering. In ICCV, 2023.
>
> [R3] Xiang Feng, Yongbo He, Yan Yang, and Yubo Wang, Yifei Chen, Zhan Wang, Zhenzhong Kuang, Feiwei Qin, Jiajun Ding. SRGS: Super-Resolution 3D Gaussian Splatting. arXiv:2404.10318, 2024.
>
> **Q1: Using GT Hand Pose.**
> A1: In the main paper, we used ground-truth pose parameters to provide an upper bound and to isolate the effect of our SR–3D reconstruction pipeline. This evaluation protocol is a widely adopted strategy in prior works [1, 2, 3, 4, 5], which also leverage GT SMPL parameters to reconstruct detailed shapes from incomplete, occluded, or degraded images while decoupling pose estimation errors from reconstruction quality.
>
> To further assess the robustness of SRHand over pose errors, we conducted experiments in the supplementary Section 3.4 by adding Gaussian noise to the GT parameters. In a multi-view video capture setup, global constraints allow pose parameters to be fitted jointly across views, reducing the joint error to *mm* scale [6, 7]. With the perturbations up to 9.54 *mm* (~1*cm*) per joint, SRHand maintained stable performance, remaining tolerant to realistic levels of pose inaccuracy. Nonetheless, extreme pose errors limit performance, as is common in most hand reconstruction pipelines.
>
> Regarding the question of the low-resolution settings and GT existness, in a multi-view body capture system, hand region resolution dynamically changes depending on its pose and camera viewpoint. Even in the same pose, one camera may capture a small region of the hand while others capture a larger region. For example, in the ActorsHQ [8] dataset, within the same pose, a single hand region varies (45×46 ~ 108×127) depending on its camera views. Thus, even though the camera captures the hand in a low resolution, global 3D pose parameters can still be extracted across multiple cameras.
>
> In addition, considering this variability observed in real datasets, we evaluated SRHand on multiple input resolutions, including 48×48, which approximates the hand region resolution in real-world datasets (ActorsHQ in Table 3 for reviewer yhNh and Supplementary Section 3.5; see also Table 1 below). We find out that as the input image resolution increases, the reconstruction accuracy typically improves. We also include experiments with even smaller hand crops to probe the lower limit of SRHand. As noted in another review (sv5s), which requested testing at extreme small scales, SRHand maintained faithful reconstruction results without significant degradation compared to other resolution scales. Nevertheless, estimating accurate hand poses from extremely low-resolution images remains an open research challenge.
>
> **Table 1:** Quantitative results of arbitrary scale input image experiment.
> | Input image resolution | Super-Resolution PSNR | Super-Resolution LPIPS | 3D Reconstruction PSNR | 3D Reconstruction LPIPS |
> |------------------------|-----------------------|------------------------|------------------------|-------------------------|
> | 32×32                  | 31.05                 | 0.0262                 | 30.23                  | 0.0341                  |
> | 48×48                  | 31.42                 | 0.0274                 | 30.43                  | 0.0334                  |
>
>
> **Q2: More experiments for 3D content generation.**
> A2: Certain works (e.g., 3DEnhancer [9] and LGM [10]) do not align with our research goal and are difficult to compare for several reasons. First, generating 3D objects is entirely different from reconstructing detailed personalized hand avatars from captured images. Reconstructing a personalized avatar is to optimize the model to capture finer details across input images, whether a generalizable 3D object generation task is to work upon a large-scale pretraining model (e.g MVDream [11]). None of the prior avatar reconstruction works [12, 13, 14] has been compared with the methods for 3D generic object generation tasks. Second, their architectures are specifically designed for static objects and scenes, not an animatable avatar. Their proposed modules, such as multi-view row attention, near-view epipolar aggregation, are designed for static objects and are not straightforward to extend for dynamic human avatars. Their training codes are also not publicly available. Regarding the existing diffusion-based models, IDM [15] and DiSR-NeRF [16] are both diffusion-based methods and are included as part of the comparative experiments in our main paper.
>
> Instead, we include experiments with SRGS [17], which learns Gaussian splatting with super-resolved images for static scenes.
> As SRGS supports a fixed upscale factor, we arbitrarily control its factor through iterative upscaling. As shown below, our method achieves performance comparable to other baselines for static 3D hand reconstruction, while still supporting animatable hand avatars. Furthermore, as the upscaling factor increases, SRHand significantly outperforms the prior works, particularly in perceptual metrics such as LPIPS and SSIM. This is because PSNR measures only the mean squared pixel difference, where overly smooth or blurred textures can still yield high scores, whereas LPIPS and SSIM better capture perceptual fidelity and structural detail.
>
>
> **Table 2:** Quantitative comparison of 3D consistency considered SR methods. NeRF-SR and DiSR-NeRF are brought from the main paper for comparisons.
> | Upscale factor | Methods     | PSNR   | LPIPS   | SSIM   |
> |----------------|-------------|--------|---------|--------|
> | ×4             | NeRF-SR     | 28.83  | 0.1410  | 0.8638 |
> |                | DiSR-NeRF   | 28.82  | 0.1303  | 0.8736 |
> |                | SRGS [17]   | **32.74** | 0.0731  | **0.9335** |
> |                | SRHand      | 29.06  | **0.0475** | 0.9016 |
> | ×8             | SRGS        | **29.22** | 0.1520  | 0.8739 |
> |                | SRHand      | 28.86  | **0.0487** | **0.9007** |
> | ×16            | SRGS        | 27.18  | 0.3872  | 0.7899 |
> |                | SRHand      | **27.52** | **0.0566** | **0.8858** |
>
> Reference:
> [1] Junying Wang, Jae Shin Yoon, Tuanfeng Y. Wang, Krishna Kumar Singh, Ulrich Neumann, Complete 3D human Reconstruction from a Singe Incomplete Image. In CVPR, 2023.
>
> [2] Jihyun Lee, Minhyuk Sung, Honggyu Choi, and Tae-Kyun Kim. Im2hands: Learning attentive implicit representation of interacting two-hand shapes. In CVPR, 2023.
>
> [3] Zeren Jiang, Chen Guo, Manuel Kaufmann, Tianjian Jiang, Julien Valentin, Otmar Hilliges, Jie Song. MultiPly: Reconstruction of Multiple People from Monocular Video in the Wild. In CVPR, 2024.
>
> [4] Minje Kim, and Tae-Kyun Kim. BiTT: Bi-directional Texture Reconstruction of Personalized Interacting Two Hands from a Single Image. In CVPR, 2024.
>
> [5] Kennard Yanting Chan, Fayao Liu, Guosheng Lin, Chuan Sheng Foo, and Weisi Lin. Robust-PIFU: Robust Pixel-Aligned Implicit Function for 3D Human Digitalization from A Single Image. In ICLR, 2025.
>
> [6] Wei Xie, Zhipeng Yu, Zimeng Zhao, Binghui Zuo, Yangang Wang. HMDO : Markerless Multi-view Hand Manipulation Capture with Deformable Objects, In Graphic Models, 2023.
>
> [7] Xiaozheng Zheng, Chao Wen, Zhou Xue, Pengfei Ren, Jingyu Wang1. HaMuCo: Hand Pose Estimation via Multiview Collaborative Self-Supervised Learning. In ICCV, 2023.
>
> [8] Mustafa Işık, Martin Rünz, Markos Georgopoulos, Taras Khakhulin, Jonathan Starck, Lourdes Agapito, Matthias Nießner. HumanRF: High-Fidelity Neural Radiance Fields for Humans in Motion. In ACM TOG, 2023.
>
> [9] Yihang Luo, Shangchen Zhou, Yushi Lan, Xingang Pan, and Chen Change Loy. 3DEnhancer: Consistent Multi-View Diffusion for 3D Enhancement. arXiv:2412.18565, 2024.
>
> [10] Jiaxiang Tang, Zhaoxi Chen, Xiaokang Chen, Tengfei Wang, Gang Zeng, and Ziwei Liu. LGM: Large Multi-View Gaussian Model for High-Resolution 3D Content Creation. In ECCV, 2024.
>
> [11] Yichun Shi, Peng Wang, Jianglong Ye, Long Mai, Kejie Li, and Xiao Yang. MVDream: Multi-view Diffusion for 3D Generation. In ICLR, 2024.
>
> [12] Tianjian Jiang, Xu Chen, Jie Song1, and Otmar Hilliges. InstantAvatar: Learning Avatars from Monocular Video in 60 Seconds. In CVPR, 2023.
>
> [13] Hezhen Hu, Zhiwen Fan, Tianhao Walter Wu, Yihan Xi, Seoyoung Lee, Georgios Pavlakos, and Zhangyang Wang. Expressive Gaussian Human Avatars from Monocular RGB Video. In NIPS, 2024.
>
> [14] Hsuan-I Ho, Jie Song, and Otmar Hilliges. SiTH: Single-view Textured Human Reconstruction with Image-Conditioned Diffusion. In CVPR, 2024.
>
> [15] Sicheng Gao, Xuhui Liu, Bohan Zeng, Sheng Xu, Yanjing Li, Xiaoyan Luo, Jianzhuang Liu, Xiantong Zhen, and Baochang Zhang. Implicit diffusion models for continuous super-resolution. In CVPR, 2023.
>
> [18]  Jie Long Lee, Chen Li, and Gim Hee Lee. Disr-nerf: Diffusion-guided view-consistent super-resolution nerf. In CVPR, 2024.
>
> [19] Xiang Feng, Yongbo He, Yan Yang, and Yubo Wang, Yifei Chen, Zhan Wang, Zhenzhong Kuang, Feiwei Qin, Jiajun Ding. SRGS: Super-Resolution 3D Gaussian Splatting. arXiv:2404.10318, 2024.

---

> > ### Comment · Reviewer_dx64 · 2025-08-05
> >
> > Thanks for the rebuttal and clarifications.
> >
> > I agree that it is fair to use GT poses as input if it's indeed widely adopted by all the other related works, even if it reduces significance of the work by making it impractical. So, it's great that you have (already had) conducted additional experiments with noise added to the pose in the supplementary - which I had missed during the review phase. As someone who is mostly interested in works which has some practical significance, I'd suggest moving that discussion into the main paper. If I were comparing works to each other to decide which method to adopt, that would be the crucial metric to check, since in practice no one can assume access to perfect GT under any setting. It would also help to add a couple of qualitative examples if there is sufficient space, as those numbers don't mean much to me. I can't tell what it means when LPIPS doubles from 0.03 to 0.06 for instance.
> >
> > Thanks for addressing all my other concerns as well. I'm inclined to increase my score as a result.

---

> > > ### Author Response · Authors · 2025-08-05
> > >
> > > Thank you very much, and we are pleased that our response has addressed the concerns. We fully acknowledge that real-world scenarios involve diverse and uncontrolled conditions where GT information is unavailable. Therefore, experiments of adding noise to GT parameters are critical for assessing robustness, which we have already explored in the supplementary material.
> > >
> > > We will revise the main paper to include these experiments and add additional qualitative results where space permits.

---

> ### Author Response · Authors · 2025-08-07
>
> Please let us know when you still have more concerns. If your concerns have been sufficiently resolved, could you kindly update your rating accordingly?

---

### Official Review · Reviewer_yhNh · 2025-07-02

**Clarity:** 2
**Significance:** 2
**Originality:** 3
**Rating:** 4
**Confidence:** 3

**Summary:**

The authors introduce SRHand, a method to reconstruct detailed 3D geometry and appearance of hands in the setting of low-resolution images. The key insight is to use an implicit image representation (GIFF) to learn prior information on hands. Experiments show superior performance in hand image super resolution. They also compared to hand avatar creation methods but not very comprehensive.

**Questions:**

- Compare with more valid baselines in 3d avatar creation
- In the final version, adjust the story and focus on 3d avatar creation.

**Ethical Concerns:**

["NO or VERY MINOR ethics concerns only"]

**Final Justification:**

My initial concerns were the lack of published baselines for comparison. The authors provided additional baselines during the rebuttal and have addressed my questions. I will raise my score but suggest the authors in the final paper to include the additional baselines, clarify research goal for the paper writing, and discuss social impact.

**Limitations:**

The authors should also discuss about the negative societal impacts. For example, it maybe a privacy concern if the avatar encodes biometric identifiers (e.g., hand appearance, geometry and motion patterns), which can affect surveillance.

**Quality:**

2

**Strengths And Weaknesses:**

## Pros

This is a very novel and practical task. Human hands are often very small in full-body images settings. Addressing the low resolution problem is very useful for large-scale reconstruction for in-the-wild videos.

Image resolution results look very impressive compared to the baselines. It contains detailed appearances such as nails and wringles.

Code will be available for research.

## Cons

**Insufficient valid baselines**: Most efforts of the experiment section is on evaluating super resolution. The evaluation is comprehensive and impressive. However, this is not the case for the main focus of the paper, hand avatar creation. Table 3 compares with UHM, and XHand but did not compare with existing baselines such as HARP [2], LiveHand [3], and Handy [4]. In particular, LiveHand also uses a super resolution method and it seems relevant to compare. Further, XHand is an arxiv and unpublished paper. It is not really a valid baseline.

**Unclear research goal**: The paper is motivated by about AR/VR and how the creation of hand avatar can help immersion (L20-22). The experiments are splitted into two main parts: hand image super-resolution and 3D hand avatar creation, where the former is emphasized a lot more than the latter. I think it is easier for the readers if the story centers around 3D hand avatar creation and use the super-resolution part as an interesting analysis. This will make the story flow a lot better.

**MipNeRF**: The paper MipNeRF [1] seems to address a similar problem on training implicit representations in low/multi-resolution settings. It seems relevant in Table 2.

Minor question: Hands are often very small in images when the full body is present but I am curious to know the source of this information - "the hand occupies less than 1.5% of captured images" (L24).

- [1] Mip-NeRF: A Multiscale Representation for Anti-Aliasing Neural Radiance Fields
- [2] HARP: Personalized Hand Reconstruction from a Monocular RGB Video
- [3] LiveHand: Real-time and Photorealistic Neural Hand Rendering
- [4] Handy: Towards a high fidelity 3D hand shape and appearance model



## Justification

Overall, I like the problem the authors are trying to address but the writing is a bit hard to follow because the story struggles with two rather different tasks. I suggest the authors to polish on this aspect. For the experiments, the 3D hand avatar method comparison is not very comprehensive.

---

> ### Author Rebuttal · Authors · 2025-07-31
>
> ## Common Response to Reviewers
> We sincerely thank the reviewers for their constructive feedback. We appreciate all reviewers recognizing the motivation of our work and the novelty of the method, along with the SOTA results. We have conducted several additional experiments to address remaining concerns.
>
> 1. 3D reconstruction experiment with other hand reconstruction methods: HARP[R1] , LiveHand [R2].
>
> 2. 3D Consistency-considered super-resolution experiment including Gaussian splatting, SRGS [R3].
>
> 3. Extreme resolution experiment to show the lower-bound of SRHand, and the results for different occlusion rates.
>
> Please refer to the additional experiment results for your consideration.
>
> References:
> [R1] Korrawe Karunratanakul, and et al. Harp: Personalized hand reconstruction from a monocular rgb video. In CVPR, 2023.
>
> [R2] Akshay Mundra, and et al. Livehand: Real-time and photorealistic neural hand rendering. In ICCV, 2023.
>
> [R3] Xiang Feng, and et al. SRGS: Super-Resolution 3D Gaussian Splatting. arXiv:2404.10318, 2024.
>
>
> **Q1: Compare with more valid baselines.**
> A1: For 3D hand reconstruction baselines, HARP [1], LiveHand [2], and other prior works (Handy [3], HandAvatar [4]) were not included as XHand [5] and UHM [6] significantly outperform these methods. Nevertheless, we present additional experiments below for HARP and LiveHand, which are among the recent works. HARP reconstructs a 3D hand using mesh subdivision with 3,093 vertices, which is not sufficient to represent a detailed hand geometric shape (significantly less than ours: 49,281 vertices). Also, assuming a one-point light condition differs from the benchmark settings used in our experiments, and yields geometric artifacts. LiveHand employs low-resolution NeRF combined with a super-resolution network for real-time hand rendering. While efficient, LR NeRF and CNN-based SR module likely loses the 3D geometric information. Both methods encounter difficulties in reconstructing 3D hands from inconsistent super-resolved images and lack performance compared to ours.
>
> **Table 1:** Quantitative comparison of 3D reconstruction results. For fair comparison, our GIIF module was used for super-resolving images.
> | SR Module | 3D Recon. Methods | PSNR   | LPIPS   | SSIM   | P2P (*mm*) |
> |-----------|------------------|--------|---------|--------|------------|
> | GIIF      | HARP [1]         | 26.25  | 0.0872  | 0.8773 | 3.38       |
> |           | LiveHand [2]     | 26.91  | 0.0821  | 0.8805 | -          |
> |           | XHand [5]    | 27.71       | 0.0507     |  0.8876    |     3.43   |
> |           | Ours             | **29.88** | **0.0362** | **0.9206** | **2.16**    |
>
>
> Mip-NeRF [7] tackles a different objective from ours; while it renders at across continuous scales, it remains bounded to the scale of training images, not performing super-resolution. Prior works of super-resolution on top of NeRF [8, 9, 10], therefore, do not directly compare with Mip-NeRF, only mentioning it as a related work.
>
> Instead, we include experiments with SRGS [11], which learns Gaussian splatting from super-resolved images for static scenes. As SRGS supports a fixed upscale factor, we arbitrarily control its factor through iterative upscaling. As shown below, our method achieves a performance comparable to other baselines designed for static scene reconstruction, while supporting dynamic and animatable hand avatars, which the other methods cannot afford to. As the upscaling factor increases, SRHand significantly outperforms the prior works, particularly in perceptual metrics such as LPIPS and SSIM. This is because PSNR measures only the mean squared pixel difference, where overly smooth or blurred textures can still yield high scores, whereas LPIPS and SSIM better capture perceptual fidelity and structural details.
>
> **Table 2:** Quantitative comparison of 3D consistency considered SR methods. NeRF-SR and DiSR-NeRF are brought from the main paper for comparisons.
> | Upscale factor | Methods     | PSNR   | LPIPS   | SSIM   |
> |----------------|-------------|--------|---------|--------|
> | ×4             | NeRF-SR     | 28.83  | 0.1410  | 0.8638 |
> |                | DiSR-NeRF   | 28.82  | 0.1303  | 0.8736 |
> |                | SRGS [11]   | **32.74** | 0.0731  | **0.9335** |
> |                | SRHand      | 29.06  | **0.0475** | 0.9016 |
> | ×8             | SRGS        | **29.22** | 0.1520  | 0.8739 |
> |                | SRHand      | 28.86  | **0.0487** | **0.9007** |
> | ×16            | SRGS        | 27.18  | 0.3872  | 0.7899 |
> |                | SRHand      | **27.52** | **0.0566** | **0.8858** |
>
>
> **Q2: Adjust the story and focus on 3d avatar creation.**
> A2: Our paper is presently written in the order of process (SR $\rightarrow$ 3D reconstruction $\rightarrow$ fine-tuning). The current structure, introducing the SR module first in the method section and presenting the SR experiments before 3D results, might overemphasize the SR module. To counteract this, we will refine the narrative and adjust the section flow to make the 3D hand avatar creation the clear focal point. Specifically:
>
> *1. Method structure (Sec. 3):* We will revise the method section to introduce SRHand as a unified 3D hand avatar reconstruction framework. The SR module (GIIF) will be described as a sub-module to support SR–3D unified structure instead of treating them as sequentially independent tasks.
>
> *2. Experiment ordering (Sec. 4):*  We will reorder the experimental results to present 3D reconstruction metrics and qualitative meshes first, followed by SR evaluations as an auxiliary task. Additionally, we also plan to balance the section length to mainly focus on the 3D avatar results and discussions.
>
> *3. Introduction (Sec 1.) & Conclusion (Sec 2.):* We will revise the introduction to clearly convey that the SR module is an integral part of the 3D reconstruction pipeline, designed to enable high-fidelity mesh recovery under low-resolution, pose-varying conditions rather than serving as an independent objective. The conclusion will be updated accordingly to emphasize SR–3D perspective as the central contribution of the work.
>
>
> **Q3: Hand Occupancy Region.**
> A3: The statement (L24) is based on our empirical observation across the full-body image datasets. We computed the pixel-wise ratio between the region using tight bounding boxes of hand and full-body in several representative samples. We present its comparisons in the table below, where a single hand occupies up to 1.5\% of the whole depending on its pose.
>
> **Table 3:** Pixel-level hand region comparison to body region across multiple datasets. (BB stands for bounding box.)
>
> | Sample # (Dataset)           | Body BB Size | Hand BB Size | Ratio  |
> |------------------------------|--------------|--------------|--------|
> | Sample 0 (Goliath [12])      | 594 × 1590   | 102 × 141    | 1.52 % |
> | Sample 1 (PeopleSnapshot [13]) | 640 × 912   | 72 × 90      | 1.11 % |
> | Sample 2 (ActorsHQ [14])     | 341 × 836    | 45 × 84      | 1.32 % |
>
>
> **Q4: Negative Social Impact.**
> A4: We agree that hand geometry and motion patterns can implicitly contain biometric cues. Our model is trained on the InterHand2.6M dataset, which is collected for public research purposes. Furthermore, the GIIF learns a generalized hand prior, not identity labels, and our 3D reconstruction pipeline cannot encode personal identifiers as long as they are absent in input images. This design setting minimizes privacy risk; still, some texture information related to personal identity can be implicitly learned in the GIIF network. We will clarify this social impact in the final version.
>
> References:
> [1] Korrawe Karunratanakul, Sergey Prokudin, Otmar Hilliges, and Siyu Tang. Harp: Personalized hand reconstruction from a monocular rgb video. In CVPR, 2023.
>
> [2] Akshay Mundra, Mallikarjun B R, Jiayi Wang, Marc Habermann, Christian Theobalt, and Mohamed Elgharib. Livehand: Real-time and photorealistic neural hand rendering. In ICCV, 2023.
>
> [3] Rolandos Alexandros Potamias, Stylianos Ploumpis, Stylianos Moschoglou, Vasilios Triantafyllou, and Stefanos Zafeiriou. Handy: Towards a high fidelity 3D hand shape and appearance model. In CVPR, 2023.
>
> [4] Xingyu Chen, Baoyuan Wang, and Heung-Yeung Shum. Hand avatar: Free-pose hand animation and rendering from monocular video. In CVPR, 2023.
>
> [5] Gyeongsik Moon, Weipeng Xu, Rohan Joshi, Chenglei Wu, and Takaaki Shiratori. Authentic hand avatar from a phone scan via universal hand model. In CVPR, 2024.
>
> [6] Qijun Gan, Zijie Zhou, and Jianke Zhu. Xhand: Real-time expressive hand avatar. arXiv:2407.21002, 2024.
>
> [7] Jonathan T. Barron, Ben Mildenhall, Matthew Tancik, Peter Hedman, Ricardo Martin-Brualla, and Pratul P. Srinivasan. Mip-NeRF: A Multiscale Representation for Anti-Aliasing Neural Radiance Fields. In CVPR, 2021.
>
> [8] Chen Wang, Xian Wu, Yuan-Chen Guo, Song-Hai Zhang, Yu-Wing Tai, and Shi-Min Hu. Nerf-sr: High-quality neural radiance fields using supersampling. In ACM, 2022.
>
> [9] Xudong Huang, Wei Li, Jie Hu, Hanting Chen, and Yunhe Wang. RefSR-NeRF: Towards High Fidelity and Super Resolution View Synthesis. In CVPR, 2023.
>
> [10] Jie Long Lee, Chen Li, and Gim Hee Lee. Disr-nerf: Diffusion-guided view-consistent super-resolution nerf. In CVPR, 2024.
>
> [11] Xiang Feng, and et al. SRGS: Super-Resolution 3D Gaussian Splatting. arXiv:2404.10318, 2024.
>
> [12] Julieta Martinez, and et al. Codec avatar studio: Paired human captures for complete, driveable, and generalizable avatars. In NeurIPS Track on Datasets and Benchmarks, 2024.
>
> [13] Thiemo Alldieck, Marcus Magnor, Weipeng Xu, Christian Theobalt and Gerard Pons-Moll. Video Based Reconstruction of 3D People Models. In CVPR, 2018.
>
> [14] Mustafa Işık, Martin Rünz, Markos Georgopoulos, Taras Khakhulin, Jonathan Starck, Lourdes Agapito, Matthias Nießner. HumanRF: High-Fidelity Neural Radiance Fields for Humans in Motion. In ACM TOG, 2023.

---

> > ### Comment · Reviewer_yhNh · 2025-08-03
> >
> > Thanks for the authors for the detailed rebuttal. After reading the author feedback as well as other reviewers' comments, I am still not very convinced by the baseline comparison.
> >
> > My main concern is the numbers reported for the baselines. For example, LiveHand reported a PSNR of 32.04 on InterHand2.6M (see Table 2 of [livehand](https://arxiv.org/pdf/2302.07672). Can the authors clarify why the numbers are different? I assume this is because of a custom evaluation set which is different from the LiveHand paper.
> >
> > I believe having the same evaluation set helps future comparison and make the results more convincing.

---

> ### Author Response · Authors · 2025-08-04
>
> If you see the table below, where 3D reconstruction methods are compared, the performances of methods vary depending on whether the input images are cropped to tightly contain hands or not. PSNR of Livehand using the whole images is 32.60, similar to what they reported. Note in Livehand [2], they use 512 x 334 whole images, where we crop the hand part to reduce the effects of black backgrounds. Prior work [1] also cropped the hand part in their experiments.
>
> **Table:** Comparison of 3D reconstruction methods using InterHand2.6M. The original high-resolution images from InterHand2.6M, not super-resolved by the proposed method, were used as input.
> |                        | Modules   | PSNR   | LPIPS   | SSIM   |
> |------------------------|-----------|--------|---------|--------|
> | **Hand crops (256 × 256)** | Livehand | 27.65  | 0.0393  | 0.8926 |
> |                        | XHand     | 28.27  | 0.0358  | 0.8953 |
> |                        | Ours      | 29.52  | 0.0349  | 0.9117 |
> | **Whole images (512 × 334)** | Livehand | 32.60  | 0.0227  | 0.9647 |
> |                        | XHand     | 33.29  | 0.0162  | 0.9670 |
> |                        | Ours      | 34.80  | 0.0101  | 0.9772 |
>
> We hope this addresses your concern.
>
>
> References:
> [1] Xingyu Chen, Baoyuan Wang, and Heung-Yeung Shum. Hand avatar: Free-pose hand animation and rendering from monocular video. In CVPR, 2023.
>
> [2] Ashay Mundra, Mallikarjun B R, Jiayi Wang, Marc Habermann, Christian Theobalt, and Mohamed Elgharib. Livehand: Real-time and photorealistic neural hand rendering. In ICCV, 2023.

---

> > ### Author Response · Authors · 2025-08-06
> >
> > Thank you again for your engagement and valuable feedback during the discussion period.
> >
> > In response to your concerns, we have provided additional experiments, including comparisons with more baselines and evaluation experiments using both hand-cropped and whole images to explain the metric score discrepancy.
> >
> > We would like to kindly ask whether our responses have sufficiently addressed your concerns. If there are any remaining issues or points that you would like us to clarify further, we would greatly appreciate your feedback and would be happy to continue the discussion.

---

> > > ### Comment · Reviewer_yhNh · 2025-08-06
> > >
> > > Thanks for the clarification. Could you report the full image numbers for all baselines (HARP, LiveHand, XHand, UHM, Ours) and metrics (PSNR, LPIPS, SSIM, P2P)? This is also important for consistent future comparison.

---

> ### Author Response · Authors · 2025-08-07
>
> As described in the main paper L226 and supplementary L17, we use 20 views x 20 frames for training, and the remaining frames are used for evaluation following prior work [5]. All compared methods (HARP, LiveHand, XHand, UHM) are learned and evaluated on our same setting using the metrics (PSNR, LPIPS, SSIM, P2P).
> Note that detailed settings vary in prior works [1, 2, 3, 4, 5], and our number of views is similar to those of LISA, XHand, and HandNeRF, where about 20 cameras are used.
>
> In case the reviewer was inquiring about the evaluation metrics of the methods with the SR module at 512x334 scale images, we report those accuracies in the table below, i.e., HARP, LiveHand, XHand, UHM, and Ours.
>
> | Modules   | PSNR  | LPIPS   | SSIM   | P2P *(mm)* |
> |-----------|-------|---------|--------|------------|
> | UHM       | 29.41 | 0.0384  | 0.8841 | 72.55      |
> | HARP      | 32.06 | 0.0286  | 0.9558 | 3.38       |
> | LiveHand  | 32.27 | 0.0275  | 0.9602 | -          |
> | XHand     | 33.12 | 0.0175  | 0.9672 | 3.43       |
> | **Ours**  | **34.28** | **0.0112** | **0.9757** | **2.16**     |
>
>
> References:
>
> [1] Akshay Mundra, Mallikarjun B R, Jiayi Wang, Marc Habermann, Christian Theobalt, and Mohamed Elgharib. Livehand: Real-time and photorealistic neural hand rendering. In ICCV, 2023.
>
> [2] Xingyu Chen, Baoyuan Wang, and Heung-Yeung Shum. Hand avatar: Free-pose hand animation and rendering from monocular video. In CVPR, 2023.
>
> [3] Z. Guo, and et al. HandNeRF: Neural Radiance Fields for Animatable Interacting Hands. In CVPR, 2023.
>
> [4] E. Corona, and et al. LISA: Learning Implicit Shape and Appearance of Hands. In CVPR, 2022.
>
> [5] Qijun Gan, Zijie Zhou, and Jianke Zhu. Xhand: Real-time expressive hand avatar. arXiv:2407.21002, 2024.

---

> > ### Comment · Reviewer_yhNh · 2025-08-07
> >
> > Thanks authors for the clarification. The rebuttal has addressed my concerns. Please incorporate all reviewer feedback to the paper should it be accepted.

---

> > > ### Author Response · Authors · 2025-08-07
> > >
> > > We thank the reviewer’s constructive feedback, and are pleased that our rebuttal has addressed the concerns. We will incorporate the suggested clarifications into the final version of the paper.

---

### Decision · Program_Chairs · 2025-09-17

**Decision:**

Accept (poster)

**Comment:**

The paper has initially received thorough reviews. The reviewers also discussed the rebuttal in depth.

The rebuttal has helped address a lot of concerns, and the reviewers agreed to revised their score.

In the current form, we recommend Accept, and encourage the authors to take into account the reviewers' comments for the final version. Congratulations!

E.g., suggestions from reviewers:
*  Include the additional baselines, clarify research goal for the paper writing, and discuss social impact.